# Transformational Leadership and Followers’ Innovative Behavior: Roles of Commitment to Change and Organizational Support for Creativity

**DOI:** 10.3390/bs13040320

**Published:** 2023-04-07

**Authors:** Kiho Jun, Joonghak Lee

**Affiliations:** 1BNU-HKBU United International College, Zhuhai 519087, China; kihojun@uic.edu.cn; 2College of Business, Gachon University, Seongnam-si 13120, Republic of Korea

**Keywords:** transformational leadership, innovative behavior, commitment to change, organizational support for creativity

## Abstract

This study aims to investigate the relationships among transformational leadership, followers’ innovative behavior, commitment to change, and organizational support for creativity. First, we examine the mediating role of commitment to change in the relationship between transformational leadership and followers’ innovative behavior, using both objective and subjective measures. Our results reveal that commitment to change indeed mediates this relationship. Second, we analyze whether the link between commitment to change and followers’ innovative behavior is moderated by organizational support for creativity. We find that this relationship is stronger for individuals with high organizational support for creativity compared to those with low support. Empirical analysis was conducted using data collected from 535 managers in 11 subsidiaries of a financial institution in South Korea. This study contributes to the management discipline by resolving mixed outcomes between transformational leadership and followers’ innovation and highlighting the influence of commitment to change and perceived organizational support for creativity on innovative behavior.

## 1. Introduction

In today’s ever-evolving business climate, innovation is viewed as a crucial factor for survival and growth [1,2,3]. For example, organizational innovation has been sparked by the COVID-19 pandemic and digital transformation to address a substantial change in the environment [4]. In addition, many scholars argue that the ability to encourage employee creativity and employee innovative behavior is a crucial factor in determining an organization’s survival and competitiveness [5,6,7,8]. Regarding how to facilitate an individual’s creativity and innovation, scholars propose that individuals’ work and social circumstances significantly influence their creativity and innovation [9]. Thus, various aspects of social environment variables have been investigated in creativity and innovation research [10]. For example, studies on creativity suggest that leadership and supervision are two of the most important contextual factors that influence creativity [11]. Additionally, the level of competition faced by firms today necessitates change-oriented leadership [12], which can foster employee innovation in a changing context [13]. Given that the ideas behind transformational leadership emphasize the transformative aspect of leaders’ actions [14], scholars explored how transformational leadership influences employee innovative behavior [15,16]. However, the question of how leadership may impact employee innovative behavior has not received the academic attention required [17,18]. Therefore, it is necessary to identify more personal and organizational factors that might influence the relationship between transformational leadership and followers’ behavior for innovation to fully understand the process.

Scholars contend that the generation of ideas or solutions, garnering of support for ideas, and implementation of an idea are three behavioral activities that comprise the multistage process of innovative behavior in the workplace [1]. As a result, innovative behavior entails the development of something novel, thereby incorporating both creative and innovative aspects [1], and it is ultimately change-oriented [19]. Furthermore, it has been asserted that innovative behavior is essentially a motivational issue as such behaviors are entirely discretionary and not formally acknowledged by organizational reward systems [20]. Thus, an essential question for leadership research and practice is how leadership can support employee innovative behavior by enhancing the employees’ motivational states.

In the current study, we propose that innovation is central to transformational leadership because leader behaviors that demonstrate respect for change should have a significant contextual impact on followers’ innovative behavior [21]. Specifically, we argue that, by utilizing four dimensions of transformational leadership, which are idealized influence, individualized consideration, inspirational motivation, and intellectual stimulation, transformational leaders can increase followers’ innovative behavior [22]. Prior research found a significant association between transformational leadership and innovation [23]. The idea that transformational leadership favorably affects creativity and innovation is theoretically supported by a few arguments. For example, transformational leaders also demonstrate unusual conduct and act as role models for innovation [24]. According to [25], a transformational leader can influence followers to re-evaluate prospective issues and their work environment, which can lead to the development of innovative ideas, by stimulating their thinking. Scholars also suggest empirical evidence for a significant relationship between transformational leadership and employee innovative behavior. For example, in studies on healthcare teams, which investigated the relationship between transformational leadership and employee innovative behavior, [26] found a relationship between transformational leadership and an environment fostering creativity, as well as between transformational leadership and implemented innovations that have been put into practice. However, the few studies that specifically looked at the relationship found mixed results e.g., [27,28,29,30]. In short, despite the abundance of conceptual work indicating a relationship between transformational leadership and employee innovative behavior, there are only a few empirical studies on this relationship. Furthermore, less definitive empirical findings suggest that there is a more nuanced relationship between transformational leadership and followers’ innovative behavior.

In the current study, we introduce followers’ commitment to change as an intermediate mechanism linking transformational leadership with followers’ innovative behavior. Furthermore, we consider organizational support for creativity as an organizational factor that influences the relationship between transformational leadership and followers’ innovative behavior.

First, we contend that a key mechanism via which transformational leadership influences employee innovative behavior in the context of transition is the commitment to change. The foundation of transformational leadership theory is the idea that this leadership style can inspire followers to think more critically [14]. Transformational leaders increase commitment to a clearly stated vision and motivate followers to adopt novel ways of thinking by appealing to their principles and beliefs. Although earlier studies showed that transformational leadership encourages creativity [31,32], these studies focused on transformational leadership in teams that functioned under largely stable circumstances. Only a few studies have touched upon the mechanisms that transformational leaders use to influence employee innovative behavior in the changing context of transition, and little is known about these processes [33]. By looking at the intermediate mechanism in the context of organizational change, our purpose is to better understand how transformational leaders boost employee innovative behavior.

Second, we consider how organizational support climate for creativity may influence the relationship between commitment to change and followers’ innovative behavior. According to organizational support theory, employees can perceive how much their organizations value them [34,35]. Employees may exhibit high levels of affective commitment, job satisfaction, and other attitudinal outcomes if they feel that the organization supports them [36,37]. Similarly, other academics have investigated the relationship between encouraging workplace culture and employee outcomes. However, research has rarely examined the effects of organizational support on leadership and organizational context [38]. Given the significance of the change context and leadership influencing creativity and innovation, organizational support for creativity can be a crucial possibility. Organizational support for creativity relates to the extent to which an employee perceives that the organization supports employees to be more creative [39]. An individual working for an organization will incur costs when they attempt to develop and implement new ideas as the organization must modify their existing system. Therefore, an employee can be innovative when they know they are backed by the organization [40].

In sum, we aimed to address the following research question using data from 535 designated change agents of one large financial institution that launched a big organizational change initiative: How does the relationship between transformational leadership and employee innovation in the context of organizational change depend on employee commitment to change, and how does organizational support for creativity moderate this relationship? Our model (see Figure 1) shows that transformational leadership raises the level of followers’ commitment to change, which then promotes followers’ innovative behavior measured by both subjective and objective indicators (i.e., self-reported innovative behavior and objective innovation performance). Moreover, the relationship between commitment to change and followers’ innovative behavior is moderated by organizational support for creativity.

By reconciling the inconsistent results of these relationships, our study contributes to the management literature. This study also responds to the topic of how innovative behavior in the changing environment can be enhanced by transformational leadership. Lastly, our paper contributes to the literature on employees’ commitment to change by revealing the mechanism of innovative behavior through organizational support for creativity using both objective and subjective indicators for innovation. Overall, this study increases our understanding of the causes and consequences of employees’ innovative behavior.

## 2. Literature Review and Hypotheses

### 2.1. Transformational Leadership and Commitment to Change

In the current study, we argue that transformational leadership is positively associated with followers’ commitment to change. Commitment to change is defined as “a force (mindset) that binds an individual to a course of action deemed necessary for the successful execution of a change initiative” [41], p. 475. According to studies, an individual’s commitment to change significantly influences employee support for change [41,42,43]. Moreover, as a theory of “change-oriented” leadership, the theory of transformational leadership implies that such leaders encourage their followers to take initiative [44]. Therefore, prior studies have highlighted the importance of transformational leadership in examining how leadership behaviors promote change [45]. Ref. [46] also hypothesized the relationship between transformational leadership and change-related outcomes by assuming that transformational leadership encourages a change among employees, which can subsequently drive change at an organizational level. Overall, followers’ commitment to change is positively associated with transformational, change-related, and communication behaviors [47].

According to scholars, managers/leaders should inspire passion and gain support from their teams for the proposed organizational transformation [48]. For example, according to studies on change, individuals who reported having a high level of trust in management and feeling that the management supports them were more receptive to suggested changes than others [49,50]. Considering the effects of leadership on change recipients’ attitudes toward change, studies on leadership have posited that charismatic or transformational leadership is particularly effective during times of change [51]. Although research has shown a strong connection between transformational leadership and employees’ commitment to the organization [51,52], data supporting the connection between transformational leadership and employees’ commitment to an organizational change are scarce [53].

We contend that transformational leaders significantly and positively influence employees’ commitment to change as change implementers due to several factors [54,55]. Firstly, transformational leaders might better combat resistance [56] as these leaders adopt “softer” influencing tactics to encourage employees to take responsibility for the change and motivate them toward self-actualization [57]. In particular, inspiring people and igniting passion entail transformational leadership, which is brought about by articulating a vision and encouraging participation in the change process [58]. Transformational leaders should be managing people’s commitment to change while simultaneously identifying and conveying a vision. Secondly, transformational leaders often use affective appeals by being upbeat and setting down the best expectations [59]. Therefore, we argue that transformational leaders are more likely to secure employees’ genuine commitment to change. Lastly, transformational leaders provide a behavioral example for subordinates to emulate that is congruent with both the values adopted by the leader and the organization’s goals [56]. Employees are more committed to supporting initiatives from leaders who lead by example and personally demonstrate what needs to happen [60].

Ref. [61] indicated that transformational leaders are more effective than less transformational leaders in situations involving transition. They specifically examined the model that showed a strong link between overall transformational leadership and the affective commitment to change of employees. Their explanation for this relationship was that these general transformational leadership behaviors create conditions in which followers are more likely to commit to a change by painting a positive vision for the future, motivating the audience, addressing their needs in a way that helps them deal with the stress and anxiety brought on by the transition, and earning their trust. In conclusion, prior research has shown that transformational leadership and employees’ commitment to change are positively related. All things considered, we hypothesized the following:

**Hypothesis** **1:**
*Transformational leadership is positively associated with followers’ commitment to change.*


### 2.2. Mediating Effect of Commitment to Change

Managers or leaders who can persuade their employees to adopt new goals, policies, and procedures may have a better probability of causing changes in organizational structures [62]. Commitment in general refers to an employee’s dedication to a goal or task [63], and commitment to change is defined as a dedication to an individual or organizational change. Thus, organizational experts have started analyzing how employees dedicated to a change work harder and more effectively to make it happen. Few studies have established the processes through which leaders’ behaviors influence outcomes in the context of organizational change. The impact of leadership on performance was mediated by the attitudes of the change beneficiaries. For instance, ref. [64] investigated the influence of leaders’ change-promoting behaviors on the effectiveness of their teams, which was mediated by the recipients’ commitment to change. In other studies, supportive leadership and perceived organizational effectiveness were mediated by employees’ attitudes toward change [65]. Although many academics investigated how change recipients’ attitudes would mediate the relationship between leadership and change-oriented results, only a few studies examined the mediating effect of employee commitment to change in the relationship between transformational leadership and employee innovative behavior [47].

To answer the preceding discussion of how followers’ innovative behavior is influenced by transformational leaders, we propose that commitment to change plays a mediating role in the connection between transformational leadership and innovative behavior. It is expected that transformational leaders will inspire and actively engage subordinates and motivate them to perform better than expected [14]. This study also highlighted existing opportunities for change and to promote follower confidence. Additionally, transformational leadership entails followers modifying their own commitments to and beliefs about organizational targets [66]. In fact, ref. [53] found that transformational leadership is effective in mobilizing people’s commitment to change. Therefore, one can similarly expect that transformational leadership will be equally effective in persuading followers to commit to organizational change.

To our knowledge, research has looked into the role that commitment to change plays as a mediator in the link between transformational leadership and innovative behavior. According to [67], self-reported innovative behavior and commitment to change are related. The innovative behavior was positively associated with charismatic leadership, and this relationship was further mediated by followers’ affective commitment to change. Ref. [68] also examined how transformational leadership encourages staff to support organizational change, which in turn sparks innovation for the organization.

In sum, we contend that transformational leadership indirectly influences employee innovative behavior by increasing the employees’ commitment to organizational change. Given that employee innovative behavior is entirely voluntary, a strong commitment to organizational transformation is essential and should increase employee excitement for innovation. As a result, we hypothesized the following:

**Hypothesis** **2:**
*Transformational leadership is positively associated with followers’ innovative behavior, mediated by followers’ commitment to change.*


## 3. Moderating Effect of Organizational Support for Creativity

As mentioned earlier, innovative behavior is a result of people being inspired by transformational leaders to be more committed to change. However, the exchange relationship quality between leaders and followers is largely ignored by the transformational leadership theory, which primarily concentrates on the role of leaders as individuals and explains how transformational leaders uplift followers’ morale and motivation [14]. A transformational leader may not be equally successful in inspiring all of their followers. We also contend that the effectiveness of transformational leadership in fostering commitment to change depends on the caliber of the interdependent connection.

According to organizational researchers, companies can successfully promote employee creativity and motivation by creating a positive work environment [69]. The literature has considered various workplace factors, including physical components, freedom, encouragement, threats, employee involvement, and knowledge sharing [70,71,72]. Although earlier studies focused on various workplace aspects, research on organizational climate is scarce [73]. Organizational climate is defined as shared perceptions of the policies, practices, and procedures employees experience, as well as the actions they see rewarded and promoted [74]. For instance, employees feel they will receive positive feedback at work and will be able to complete their tasks on their own when they perceive organizational support [75]. Therefore, it is necessary to look into how organizational support may affect the larger social milieu. Importantly, businesses that operate in an environment that is undergoing rapid change must maintain their competitive advantage by encouraging employee innovation to produce new services or goods [76].

As organizational change is a dynamic process, each employee’s commitment to change must be supported by various motivations [77]. Perceived organizational support can meet socioemotional needs, strengthen organizational commitment, and promote improved psychological wellbeing [38]. Positivity for the organization, such as work engagement and organizational citizenship behavior, can result from a higher level of perceived organizational support [78]. Notably, a meta-analytic review by [36] revealed that perceived organizational support is associated with task performance.

Task performance is one of the most crucial outcomes achieved in the management discipline [79]. It refers to meeting formal job requirements that follow prescribed work procedures, a type of in-role behavior, and routine administrative work [80]. However, creativity is an extra role and discretionary behavior that promotes innovation [5,81]. Since creativity is a vital characteristic for firms to set themselves apart from their competitors [82], organizations should encourage their employees to nurture creativity through an organizational culture and environment that rewards innovative behavior [83]. Moreover, it has long been believed that creativity is a major factor in innovation [84]. Creativity and innovation are perceived to go hand in hand [85].

Creativity has traditionally been characterized as the invention of new and useful ideas, as opposed to innovation, which has traditionally been defined as the generation of original and useful ideas as the first step, with their application as the second stage [5,86]. Several studies have revealed how crucial organizational support for creativity is for innovation [84,85,86,87]. The relationship between organizational support and innovation performance was empirically tested in studies [88]. However, not many studies explored this relationship [89], and earlier studies took an abstract approach to the subject [83]. Together, our paper explores the relationship between commitment to change and both objective and subjective innovation performance by examining the effects of organizational support for creativity on this relationship. Hence, we hypothesize the following:

**Hypothesis** **3:**
*Organizational support for creativity moderates the relationship between followers’ commitment to change and followers’ innovative behavior.*


## 4. Moderated Mediating Effect

According to Hypothesis 2, commitment to change has a mediating role in the relationship between transformational leadership and employee innovative behavior. Additionally, Hypothesis 3 illustrates the moderating effect of organizational support for creativity on the said relationship. According to Hypotheses 1–3, organizational support for creativity has a moderating influence on the effect of transformational leadership on employee innovative behavior. This moderating effect can be seen as a moderated mediation model [90]. Combining these hypotheses, we suggest that the relationship between transformational leadership and employee innovation is more effectively mediated by organizational support for creativity than by a commitment to change alone.

By articulating short- and long-term visions, leaders can define and shape the workplace where employees are able to define goals and discover solutions together [23]. Accordingly, transformational leaders can intrinsically motivate employees and are positively related to innovative behavior [31]. Employees who strongly believe that their businesses adequately support their creativity are, therefore, more likely to feel at ease participating in the firm’s innovation programs and exhibit more innovative behaviors [91], which leads to improved innovation performance. Therefore, the indirect impact of transformational leadership on staff creativity should be more significant. As a result, we suggest the following hypothesis:

**Hypothesis** **4:**
*The indirect effect of transformational leadership on followers’ innovative behavior, via followers’ commitment to change, is moderated by organizational support for creativity such that the indirect effect is stronger when organizational support for creativity is high.*


## 5. Methods

### 5.1. Research Setting

We selected one of Korea’s largest financial organizations, the site of a significant organizational change program, as our research location. A sample from one of Korea’s largest financial organizations that underwent a significant organizational change program should provide an excellent opportunity to investigate the relationship among transformational leadership, employee commitment to change, and employee innovative behavior. In other words, investigating employees’ innovative behaviors in such an innovative environment would provide valuable insights into the relationship among transformational leadership, employee commitment to change, and employee innovative behavior in the context of organizational change.

### 5.2. Sample and Procedure

From the financial group’s 11 subsidiaries where a company-wide change program was launched, 535 middle-managers completed the study’s survey online. We used a self-report research design except for followers’ innovative performance; therefore, there was an interval of approximately 1 month between two rounds of the survey. Participants were questioned in the initial phase regarding the transformational leadership style, level of commitment to change, and other demographic details of their supervisors (Timepoint 1). In the second phase, they assessed the level of their innovative behavior over the previous weeks (Timepoint 2). In the first phase, a total of 560 responses were collected (from 700 managers). In the second phase, we obtained 535 valid and completed questionnaires (from 560 managers) (the final response rate was 76.4%). Of the respondents, 77% were male. In terms of rank, 12% were assistant managers, 35% were managers, 37% were deputy general managers, 1.2% were general managers, and 14% were assistant branch managers. Regarding their tenure, 43% had held their positions for more than 20 years, with 6.2% for less than 3 years, 16% for more than 3 years but less than 10 years, and 34% for more than 10 years but less than 20 years.

### 5.3. Measures

Transformational leadership. We used items from the Multifactor Leadership Questionnaire to assess transformative leadership (MLQ Form 5X) [22]. Using these items, all respondents assessed their direct supervisor’s leadership style. After in-depth discussions with the HR managers, we determined that the eight items for charisma (idealized influence), four items for inspirational motivation, and four items for intellectual stimulation were the most relevant to our study. We aggregated these three subcomponents into an overall measure of transformational leadership, as performed in previous studies (Cronbach’s alpha = 0.97).

Organizational support for creativity. Using four questions developed by [1], we measured organizational support for creativity. These items were used by each change agent to assess how they felt about organizational support for creativity. A sample item is “our ability to function creatively is respected by the leadership” (Cronbach’s alpha = 0.86).

Commitment to change. We utilized four items developed by [92] to assess commitment to change. Using these items, each responder evaluated the extent of their commitment to change. A sample item is “I am doing whatever I can to help this change be successful” (Cronbach’s alpha = 0.77).

Innovative behavior. Two different measures were used to assess followers’ innovative behavior of the workforce. This study differs from others as it focuses on employee innovative behavior in addition to creativity (e.g., idea generation and idea implementation). Given that the bulk of the predictors were also self-report measures [93], a recent review indicated that self-reported measures of creativity tended to have greater impact sizes with predictors [94]. Additionally, supervisory ratings are used as an outcome metric for innovative behavior in studies of an individual [95]. As a result, scholars contend that having more varied sources of outcome measures would allow researchers to be more confident about their findings. Researchers advise that, to cross-validate self-reported ratings [95], future studies should make an effort to gather a second measure of innovative behavior (e.g., objective indices in our study). In line with this advice, we assessed followers’ innovative behavior with both subjective and objective innovation indicators: self-reported innovative behavior and innovation performance. The usage of two measures is essential for obtaining a comprehensive understanding of followers’ innovative behavior. By incorporating both self-reported and performance measures, we can address potential limitations and biases inherent to each approach, ultimately providing a more robust assessment of the relationships under investigation.

First, we measured followers’ innovative behavior with self-reported measures. Specifically, self-reported innovative behavior was measured using six items developed by [1]. By responding to survey questions, each responder assessed their inventive activity. A sample item is “I develop adequate plans and schedules for the implementation of new ideas” (Cronbach’s alpha = 0.83). Second, we employed objective measures to assess followers’ innovative behavior because employee innovation incorporates both the generation of novel ideas (i.e., creativity) and their implementation (converting these ideas into new and improved products or services); we measured followers’ innovative behavior with an objective measure provided by the company, i.e., the “mileage” accumulated (in dollars) by each individual in providing new ideas that are implemented. Each employee offers fresh concepts or recommendations via the company website, after which such submissions are reviewed by special committee members. After evaluations, the committee decides regarding the suitability of new concepts for implementation in the workplace. The employees receive compensation for their ideas if they are put into practice at work. In light of this, we believe that the “mileage” gained through this process can reflect all facets of innovation: idea generation, idea championing, and idea implementation.

*Control variables.* Past research consistently linked a few demographic factors to creative habits e.g., [96,97]. Therefore, we controlled for the demographic variables formal organizational rank, tenure, age, gender, and subsidiaries to rule out the likelihood that the results were the consequence of a spurious connection. We added subsidiaries as controls because respondents came from 11 different subsidiaries of this company group. Additionally, we managed the time spent working with their supervisor and longevity in the team, both of which could have impacted the dependent variable.

### 5.4. Analytical Strategy

In order to test the proposed hypotheses, we utilized SPSS 25 software with PROCESS macro and conducted a bootstrapping analysis [98,99]. Furthermore, we also examined the moderating effect by visualizing the simple slope [100].

## 6. Results

### 6.1. Confirmatory Factor Analysis

To determine the fitness of the proposed model, we first performed a series of confirmatory factor analyses (CFAs). The CFAs considered all four constructs (transformational leadership, commitment to change, organizational support for creativity, and followers’ innovative behavior). We deleted one item (Support 3) as it did not suit the survey’s context very well. As all the remaining loadings were higher than the suggested threshold, convergence validity was attained. The results of the proposed three-factor structure illustrated in Table 1 demonstrated a good fit with data (χ^2^ = 691.67, df = 279, χ^2^/df = 2.48, CFI = 0.964, IFI = 0.964, TLI = 0.958, and RMSEA = 0.053). We examined a two-factor model that combined organizational identity and affective commitment into a single factor in comparison to this three-factor baseline model. The fit indices back up the suggested three-factor model, as shown in Table 1. We provide the results without adjustments of presentational parsimony, although we do provide their bivariate correlations with the study variables in Table 1 [101].

### 6.2. Descriptive Statistics

Table 2 presents the descriptive statistics, correlations, and reliability for all variables. As hypothesized, transformational leadership had a substantial positive correlation with followers’ innovative behavior (r = 0.21, *p* < 0.01). However, it had no relationship with innovation performance. Additionally, it was discovered that organizational support for creativity (r = 0.15, *p* < 0.01) and commitment to change (r = 0.46, *p* < 0.01) were both positively correlated with transformational leadership.

### 6.3. Hypothesis Testing

Our model-based hypotheses stated that Hypothesis 1 would show a positive relationship between transformational leadership and followers’ commitment to change. We applied ordinary least squares regression to test this hypothesis. Transformational leadership was positively associated with followers’ commitment to change, according to ordinary least squares regression analysis (β = 0.38, *p* < 0.01, see Table 3). This suggests that employees who perceived their leaders as exhibiting transformational leadership behavior were more likely to be committed to organizational change.

Second, Hypothesis 2 asserted that enhanced commitment to change serves as a mediator between transformational leadership and is positively associated with followers’ innovative behavior as measured with two indicators (i.e., self-reported innovative behavior and innovation performance). Using the PROCESS macro, we performed a bootstrapping-based mediation test to examine Hypothesis 2 [98]. We calculated a 95% confidence interval (CI) around the estimated indirect impacts of transformational leadership on followers’ innovative behavior through commitment to change using unstandardized coefficients and a bootstrapping procedure with 5000 resamples [102]. The bias-corrected 95% CI must exclude 0 for the bootstrapped indirect effect to be considered significant. We used two indicators to assess followers’ innovative behavior, as stated in the section on measurements. First, the results showed that, through the influence of commitment to change, transformational leadership was linked to an increase in followers’ self-reported innovative behavior among employees (indirect effect = 0.15, SE = 0.02, 95% CI = 0.11 to 0.20). Second, for innovation performance, we discovered that transformational leadership was indirectly related to increased innovation performance, mediated by commitment to change (indirect effect = 6.15, SE = 2.97, 95% CI = 0.39 to 12.29). These findings collectively support Hypotheses 1 and 2 (see Table 3 and Table 4).

Hypothesis 3 proposes that the relationship between commitment to change and followers’ innovative behavior is moderated by organizational support for creativity. First, model 2 in Table 5 demonstrates a positive and substantial interaction between commitment to change and organizational support for creativity on follower’s innovative behavior (β = 0.11, *p* < 0.01). Given conditional values of organizational support for creativity, we rearranged the multiple regression equation into simple regressions to interpret this moderating impact [100]. This relationship is plotted in Figure 2, which shows that commitment to change was related to innovative behavior for employees with high organizational support for creativity. In contrast, the flat slope shows that commitment to change did not affect followers’ innovative behavior for individuals with low organizational support for creativity; commitment to change was associated with innovative behavior. Second, model 3 in Table 6 demonstrates that the interaction between commitment to change and organizational support for creativity on innovation performance is also significant (β = 17.67, *p* < 0.05), as shown in Figure 3. Plots of the interaction terms in Figure 3 demonstrate that commitment to change was related to innovative performance for individuals with high organizational support for creativity. These analyses collectively support Hypothesis 3, which suggests that organizational support for creativity moderates the relationship between commitment to change and followers’ innovative behavior and innovative performance.

Lastly, we used the PROCESS macro in SPSS 25 to analyze Hypothesis 4 [103]; Table 7 and Table 8. For innovative behavior as a dependent variable, according to the findings from 5000 resamples, the indirect effect of transformational leadership on innovative behavior was statistically significant at high levels of moderation (conditional indirect effect = 0.19, SE = 0.03, 95% CI = 0.14 to 0.25) but weakened at low levels (conditional indirect effect = 0.12, SE = 0.03, 95% CI = 0.07 to 0.18). To determine the statistical significance of the moderated mediation effect, we developed the index of moderated mediation. The index was significant (Index = 0.04, SE = 0.17, 95% CI = 0.01 to 0.07). Second, the findings for employee innovation performance suggest that the indirect effect of transformational leadership on innovation performance was significant at high levels of moderation (conditional indirect effect = 11.26, SE = 3.47, 95% CI = 5.36 to 18.87) but not at low levels (conditional indirect effect = 1.00, SE = 4.45, 95% CI = −7.96 to 9.56). The index of moderated mediation was also significant (Index = 0.05, SE = 0.02, 95% CI = 0.01 to 0.08). Consequently, these findings support Hypothesis 4. Our general moderated mediation hypothesis (Hypothesis 4) is supported by these findings, indicating that the relationship among transformational leadership, employee commitment to change, and followers’ innovative behavior and innovation performance is moderated by organizational support for creativity (Figure 4).

## 7. Discussion

This study sought to address the following unanswered questions: How do transformational leaders enhance followers’ innovative behavior? Under which conditions is transformational leadership more effective to increase followers’ innovative behavior? Using data collected from 535 middle-managers in one financial institution in Korea, first, we found that followers’ perception of transformational leadership was positively associated with their commitment to change. Second, we found that transformational leadership influenced followers’ innovative behavior via increased commitment to change. Lastly, we also found that the relationship between commitment to change and followers’ innovative behavior was moderated by perceived organizational support for creativity.

### 7.1. Theoretical Implications

Our research contributes several theoretical insights to the existing knowledge on leadership and innovation. Firstly, in line with previous research findings, we demonstrated that transformational leadership positively influences employees’ commitment to organizational change [50]. Our study extends the investigation of the relationship between transformational leadership and commitment to a new domain: organizational change and employee innovation during times of change. Consequently, transformational leaders appear to enhance employee commitment to organizational change, thereby fostering employee innovation [67].

The second contribution is the identification of a psychological mechanism through which transformational leadership promotes employee innovative behavior. Our results suggest that transformational leaders act as facilitators of their followers’ innovative behavior [104]. Furthermore, we found that employees’ willingness to embrace change serves as a critical mediating variable in the relationship between transformational leadership and followers’ innovative behavior. Although the association between transformational leadership and employee innovation has been extensively theorized and empirically tested [51], prior results were inconclusive, and only a few studies focused on the impact of transformational leadership behaviors on employee innovation [47,105]. By examining the mechanisms through which transformational leadership fosters employee innovative behavior via change recipients’ responses, our research advances the understanding of transformational leadership’s role [106]. Our study is notable for being among the first to establish the positive relationship between transformational leadership and employee innovation and to identify the mediating factors connecting transformational leadership to employees’ willingness to engage in innovation activities.

Thirdly, our research enriches the leadership literature by elucidating the connection between transformational leadership and change recipients’ attitudes toward organizational change, shedding light on the moderating role of organizational support for creativity. Our findings suggest that the quality of employee–supervisor relationships influence the extent to which transformational leadership affects change recipients’ reactions to organizational change [39]. In other words, transformational leaders who establish strong relationships with their subordinates in the context of organizational change exert a more significant influence on employees’ attitudes toward that change [47].

The fourth contribution is the evaluation of our approach within a real business scenario where the company embarked on a significant reform initiative to ensure future survival. Most prior research focused on transformational and transactional leadership in units operating under relatively stable conditions [105]. Our study adds value to the leadership and change literature, as neither adequately addresses the specific relevance of leader behavior attributes in the context of organizational change or the unique role of leaders during change [106]. This allows for the enhancement of employee innovation through transformational leadership within the context of organizational change. Moreover, the utilization of both self-reported and performance-based measures is crucial for capturing a comprehensive view of followers’ innovative behavior and ensuring that the findings are valid, reliable, and applicable across various contexts. This approach enables a deeper understanding of the intricate relationships among transformational leadership, commitment to change, and organizational support for creativity in promoting innovation.

### 7.2. Managerial Implications

Our study presents several managerial implications. Firstly, the mediating effect of an employee’s commitment to change elucidates the manner in which transformational leadership influences followers’ innovation. From a leadership perspective, leaders should be more cognizant of how their actions impact employees’ commitment to change, which serves as a crucial attitudinal predictor of innovation [67]. Organizations should consider leveraging transformational leadership to enhance employee innovation by addressing both contextual and personal factors [50]. For example, a training program could be developed for executives to promote behaviors that foster employees’ positive attitudes toward change, subsequently leading to innovation [107].

Secondly, our research proposes a more practical strategy for boosting innovation. Employee commitment to change, resulting from transformational leadership, is regarded as an internal driver of innovation, contrasting with traditional approaches that stimulate creativity through external rewards. Organizations often conserve resources during times of change to prepare for unforeseen uncertainties that may affect their operations [108]. Moreover, a 40-year meta-analysis indicated that intrinsic motivation tends to exert a stronger influence on employees’ attitudes and performance [109]. Thus, to stimulate innovative behaviors among employees, practitioners should select and cultivate transformational leaders.

Lastly, the moderating effect of organizational support for creativity found in our study has implications for enhancing the relationship between commitment to change and employee innovation. Employees’ positive attitudes and actions are significantly influenced by workplace culture [110]. An employee’s performance may vary on the basis of the degree of encouragement within the workplace culture [111,112]. Our study highlights that the organizational climate can strengthen the link between employees’ commitment to change and their innovativeness, particularly focusing on the support that organizations provide for creativity. Consequently, managers and employees should recognize the impact of organizational support for creativity and establish procedures and guidelines to create a supportive work environment.

### 7.3. Limitations, Directions for Future Research, and Proposals

Despite the contributions made by our study, it also had some limitations. Its cross-sectional research design is a clear weakness that made it challenging to draw firm conclusions about the causality. Although the results appear to support the hypotheses, caution must be used when determining the causation of the findings. For instance, creative individuals could be able to share more ideas with their management, leading the employee to perceive the manager as a transformational leader. The second limitation is related to potential issues with common method variance (CMV). All constructs (except the innovation performance measure) were self-reported by the respondents, thereby increasing the likelihood that observed connections would have different results due to common method bias [93]. Therefore, we tried to reduce sampling bias and CMV. In the survey design, we were careful to adhere to the recommendations of [93] by separating the questions used in the research. When analyzing perceptual outcomes and internal states, such as feelings and perceptions, self-report data are widely regarded as the most valid method, which is relevant to CMV [113]. In conclusion, we contend that the self-reported data in our study would not be a severe issue given that some questions in our study were related to the employees’ own perceptions. The third limitation is that we only used a sample of South Korean workers to test our hypotheses; thus, we may be unable to confidently generalize our findings [114]. Hence, future research must replicate our model using various samples from different countries.

On the basis of the findings of our study, we suggest several avenues for future research. First, scholars could examine the effectiveness of other leadership styles, such as transactional or authentic leadership, in fostering employee innovative behavior in the context of organizational change. Second, researchers could explore the role of other individual-level factors, such as psychological capital or emotional intelligence, in mediating the relationship between leadership and employee innovative behavior. Third, future studies could investigate the moderating role of other contextual factors, such as organizational culture or technological advancement, in the relationship between leadership and employee innovative behavior.

## 8. Conclusions

Our research emphasizes the critical role that leadership plays in fostering employee innovative behavior by influencing workers’ attitudes. Our study sheds new light on the role of leadership in the process by which leaders foster employee innovative behavior, given that the dynamic and competitive environment forces organizations to increase their flexibility, responsiveness, and efficiency, leading to continued innovation. With a focus on the mediating role of employees’ commitment to change and the moderating role of organizational support for creativity, we believe this study is one of the first attempts to provide empirical evidence demonstrating the mechanism via which transformational leaders facilitate employees’ innovative behavior.

## Figures and Tables

**Figure 1 behavsci-13-00320-f001:**
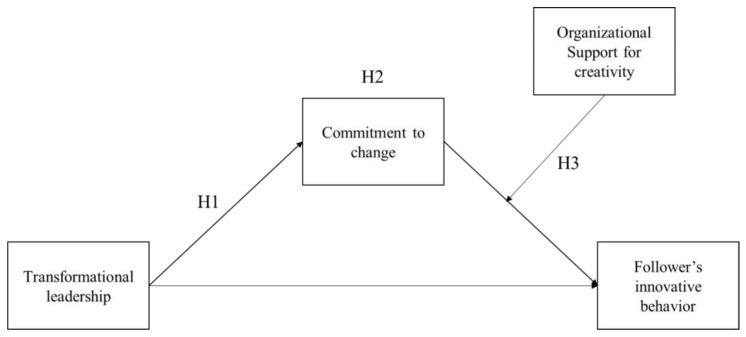
Theoretical research model (made by authors, 2023).

**Figure 2 behavsci-13-00320-f002:**
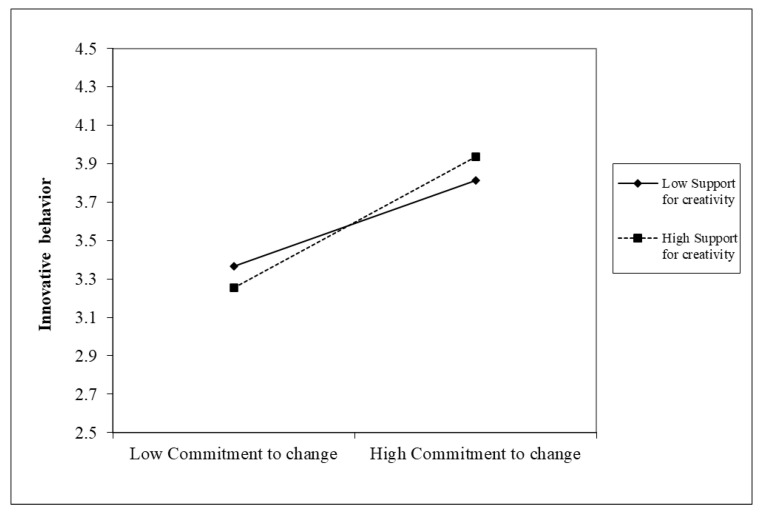
Interaction effects of support for creativity on innovative behavior (self-reported).

**Figure 3 behavsci-13-00320-f003:**
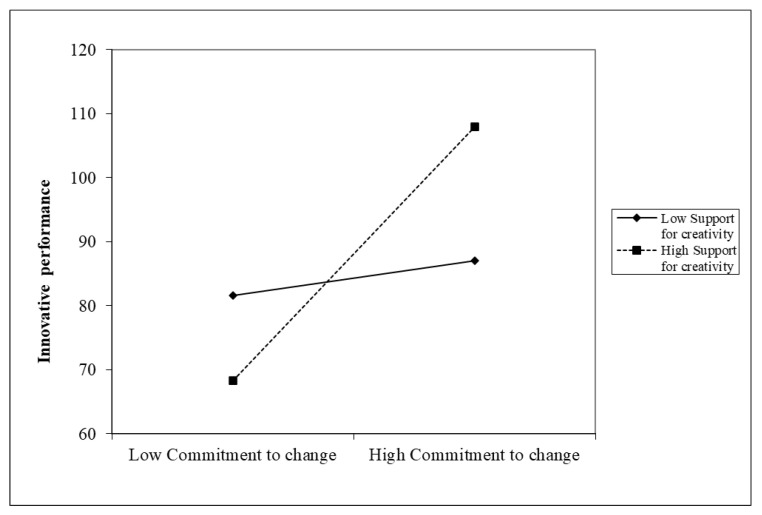
Interaction effects of support for creativity on innovative behavior (performance).

**Figure 4 behavsci-13-00320-f004:**
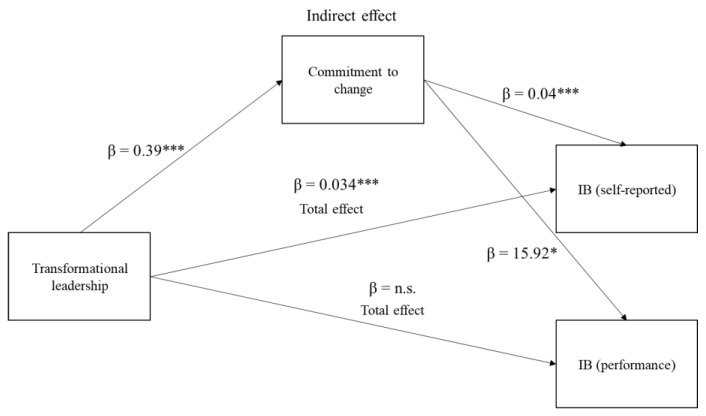
Results summarizing empirical analysis (made by authors, 2023). * *p* < 0.05, *** *p* < 0.001, n.s.; not significant.

**Table 1 behavsci-13-00320-t001:** Measurement model and results of confirmatory factor analysis.

Model	c^2^	df	v^2^/df	RMSEA	NFI	IFI	TLI	CFI
Four-factor model	112.89	50	2.26	0.07	0.95	0.97	0.96	0.97
Three-factor model	184.79	51	3.62	0.10	0.98	0.94	0.92	0.94

Note. df = degrees of freedom, RMSEA = root-mean-square error of approximation, NFI = normed fit index, IFI = incremental fit index, TLI = Tucker–Lewis index, CFI = comparative fit index.

**Table 2 behavsci-13-00320-t002:** Descriptive statistics, correlations, and reliability.

	M	SD.	1	2	3	4	5	6	7	8	9	10	11
1.Rank	2.63	1.13											
2.Age	41.47	5.46	0.78 **										
3.Gender	1.22	0.42	−0.23 **	−0.19 **									
4.Sub.	4.02	2.18	−0.32 **	−0.27 **	−0.04								
5.Tenure	3.11	0.90	0.54 **	0.69 **	0.12 **	−0.42 **							
6.TinT	2.18	1.11	−0.08	−0.05	−0.06	0.20 **	−0.07						
7.PW	1.69	0.94	−0.13 **	−0.11 *	−0.03	0.28 **	−0.12 **	0.42 **					
8.TL	3.80	0.73	0.13 **	0.10 *	−0.01	−0.07	0.12 **	−0.04	−0.02	(0.97)			
9.OSC	3.87	0.68	0.10 *	0.08	−0.08	0.13 **	0.02	0.03	0.13 **	0.42 **	(0.86)		
10.CC	3.30	0.71	0.10 *	0.12 **	0.05	0.02	0.14 **	0.05	0.01	0.45 **	0.41 **	(0.77)	
11.IB	3.54	0.61	0.15 **	0.10 *	−0.15 **	−0.02	−0.03	0.06	0.10 *	0.21 **	0.46 **	0.18 **	(0.83)
12.IP	78.19	105.31	−0.01	−0.04	−0.12 **	0.18 **	−0.10 *	0.15 **	0.15 **	0.05	0.15 **	0.07	0.05

Note. N = 535. * *p* < 0.05, ** *p* < 0.01; SubTinT: tenure in team, PW: period working with the supervisor, TL: transformational leadership, OSC: organizational support for creativity, CC: commitment to change, IB: innovative behavior, IP: innovation performance.

**Table 3 behavsci-13-00320-t003:** Regression analysis for mediation analysis.

Var.	DV = CC					DV = IB (Self-Reported)					DV = IB (Performance)				
	B	SE	t	LLCL	ULCL	B	SE	t	LLCL	ULCL	B	SE	T	LLCL	ULCL
Con.	1.90	0.32	5.85	1.26	2.54	2.00	0.29	6.78	1.42	2.58	17.39	57.11	0.30	−94.80	129.6
TL	0.38	0.03	10.80	0.31	0.45	0.01	0.03	0.46	−0.05	0.08	3.03	6.73	0.45	−10.19	16.26
CC						0.39	0.03	10.37	0.32	0.47	15.92	7.42	2.14	1.34	30.49

Note. N = 535. Var.: variables, Con.: constant, TL: transformational leadership, CC: commitment to change, IB: innovative behavior.

**Table 4 behavsci-13-00320-t004:** Summary of indirect effects in the relationship among TL, CC, and IB.

Indirect Effect	B	Boot SE	Boot LLCL	Boot ULCL
TL -> CC -> IB (self-reported)	0.15	0.02	0.11	0.20
TL -> CC -> IB (performance)	6.15	2.97	0.39	12.29

Note. N = 535. TL: transformational leadership, CC: commitment to change, IB: innovative behavior.

**Table 5 behavsci-13-00320-t005:** Regression analyses of IB (self-reported) for the mediator and moderator.

Var.	Mediator = CC						DV = IB (Self-Reported)					
	B	SE	t	*p*	LLCL	ULCL	B	SE	t	*p*	LLCL	ULCL
Con.	1.90	0.32	5.85	0.00	1.26	2.54	3.49	0.59	5.88	0.00	2.32	4.66
TL	0.38	0.03	10.80	0.00	0.31	0.45	0.01	0.03	0.40	0.68	−0.05	0.08
CC							0.02	0.13	0.14	0.88	−0.25	0.29
OSC							−0.46	0.16	−2.81	0.00	−0.78	−0.14
CC × OSC							0.11	0.04	2.88	0.00	0.03	0.19

Note. N = 535. OSC: organizational support for creativity, CC: commitment to change, IB: innovative behavior.

**Table 6 behavsci-13-00320-t006:** Regression analyses of IB (performance) for the mediator and moderator.

Var.	Mediator = CC						DV = IB (Performance)					
	B	SE	t	*p*	LLCL	ULCL	B	SE	t	*p*	LLCL	ULCL
Con.	1.90	0.32	5.85	0.00	1.26	2.54	238.21	115.00	2.07	0.03	12.28	464.13
TL	0.38	0.03	10.80	0.00	0.31	0.45	1.77	7.12	0.24	0.80	−12.22	15.76
CC							−41.57	26.58	−1.56	0.11	−93.79	10.65
OSC							−65.67	31.81	−2.06	0.04	−128.17	−3.17
CC × OSC							17.67	7.94	2.22	0.02	2.06	33.28

Note. N = 535. OSC: organizational support for creativity, CC: commitment to change, IB: innovative behavior.

**Table 7 behavsci-13-00320-t007:** Summary of moderated mediation effects for innovative behavior (self-reported).

Index of Moderated Mediation		Index	Boot SE	Boot LLCL	Boot ULCL
OSC		0.04	0.17	0.01	0.07
Indirect effect	OSC	B	Boot SE	Boot LLCL	Boot ULCL
TL -> CC -> IB (self-reported)	2.500	0.122	0.026	0.074	0.176
	3.250	0.157	0.024	0.111	0.207
	4.000	0.191	0.029	0.136	0.249

Note. N = 535. TL: transformational leadership, CC: commitment to change, IB: innovative behavior.

**Table 8 behavsci-13-00320-t008:** Summary of moderated mediation effects for innovative behavior (performance).

Index of Moderated Mediation		Index	Boot SE	Boot LLCL	Boot ULCL
OSC		0.046	0.017	0.012	0.079
Indirect effect	OSC	B	Boot SE	Boot LLCL	Boot ULCL
TL -> CC -> IB (performance)	2.500	1.008	4.452	−7.962	9.564
	3.250	6.134	3.072	0.384	12.513
	4.000	11.261	3.473	5.357	18.872

Note. N = 535. TL: transformational leadership, CC: commitment to change, IB: innovative behavior.

## Data Availability

Data is unavailable due to privacy or ethical restrictions.

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
