# Peer review of "Transformational Leadership and Followers’ Innovative Behavior: Roles of Commitment to Change and Organizational Support for Creativity"

_behavsci, 2023, doi:10.3390/bs13040320_

Round 1

Reviewer 1 Report

Dear authors,

Thank you for your paper - it is very interesting, and maybe my comments can improve it.

1. Please check the journal referencing rules

2. Please check the grammar and writing for potential typos - line 29 Innovation 

3. Make sure that the Figure and its title are on the same page

4. Remove the page numbers from the footer

5. Why was the company selected? 

6. Is the second phase number 535 out of first phase 560 or?

7. The point that there was compensation to the respondent can put in question your findings -  i.e., they filled out the survey because they were paid. How can you be sure that the findings are good and not that I wanted the money, so I filled out the survey? This, for me, represents the problem, and you didn't state anywhere in the paper who sponsored or funded your research.

8. Why is everything in BOLD from line 503 to line 570?

9. Table 2 is out of the page limits - please correct it

10. It would be nice to see proposals for future research in the conclusion and maybe to make shorter sentences since the sentences in 5-6 rows are too long and hard to understand.

Author Response

You can post and reply to comments about the manuscript here. Note that the reviewers can also read these comments.

 : We want to first thank you and the Reviewer for providing us with the opportunity to revise and resubmit our research. It is clear that you and the Reviewers gave this manuscript (2277882) very careful attention, and we have tried to reciprocate by addressing all of the various comments to the fullest extent possible. Below we have reproduced reviewers’ feedback in bold italics, followed by our response (blue color) in regular font. We carefully read the three reviewers’ comments and addressed and/or responded to each comment. We greatly appreciate this opportunity to revise and resubmit our manuscript. We welcome your continued feedback as we strive to make our manuscript as helpful to the field as possible.

Reviewer: 1

Recommendation:

Thank you for your paper - it is very interesting, and maybe my comments can improve it.

Comments:

  1. Please check the journal referencing rules.

Response to reviewer: Thanks for your feedback. We have carefully checked the referencing rules of the journal and made sure that all the references are in the correct format.

  1. Please check the grammar and writing for potential typos - line 29 Innovation.

Response to reviewer: Thanks for your feedback. We have carefully checked the manuscript for grammar and potential typos, and corrected the error on line 29 regarding "Innovation".

  1. Make sure that the Figure and its title are on the same page.

Response to reviewer: Thanks for your feedback. We have ensured that all figures and their titles are on the same page to improve readability and clarity.

  1. Remove the page numbers from the footer.

Response to reviewer: Thanks for your feedback. We have removed the page numbers from the footer to comply with the journal's formatting guidelines.

  1. Why was the company selected? 

Response to reviewer: We appreciate your insightful comments and suggestions for improvement. Regarding your question on why we selected the sample from one of Korea's largest financial organizations, we would like to clarify that we chose this organization due to its significant organizational change program, which provided an excellent opportunity to investigate the relationship between transformational leadership, employee commitment to change, and employee innovation in the context of organizational change.

As we mentioned in research setting section of our revised manuscript , this company's OneDo innovation initiative was launched in response to the global financial crisis to transform the group into a low-cost, highly efficient, and innovative organization. Moreover, the program was implemented using a bottom-up approach, with employees participating voluntarily in its activities. Thus, studying the employees' innovative behaviors in such an innovative environment would provide valuable insights into the relationship between transformational leadership, employee commitment to change, and employee innovation in the context of organizational change.

We hope this explanation addresses your concern adequately. Once again, we appreciate your thoughtful feedback, and we have made the necessary revisions to our manuscript accordingly.

  1. Is the second phase number 535 out of first phase 560 or? 

Response to reviewer: Thanks. We have clarified that the second phase numbered 535 is out of the first phase, which numbered 560.

  1. The point that there was compensation to the respondent can put in question your findings -  i.e., they filled out the survey because they were paid. How can you be sure that the findings are good and not that I wanted the money, so I filled out the survey? This, for me, represents the problem, and you didn't state anywhere in the paper who sponsored or funded your research.

Response to reviewer: Thank you for your comments regarding the compensation provided to survey respondents. We have double-checked with the survey company and confirmed that there was no compensation given, and we have removed any mention of it from the paper. Instead, the company actively encouraged participation, which we believe resulted in a higher response rate. We have updated the paper accordingly and believe this strengthens the study.

  1. Why is everything in BOLD from line 503 to line 570?

Response to reviewer: Thanks for your feedback. The text from line 503 to line 570 has been modified to remove the bold formatting that was mistakenly applied.

  1. Table 2 is out of the page limits - please correct it

Response to reviewer: Thanks for your feedback. We have corrected Table 2 to ensure that it falls within the page limits of the journal.

  1. It would be nice to see proposals for future research in the conclusion and maybe to make shorter sentences since the sentences in 5-6 rows are too long and hard to understand.

Response to reviewer: Thank you for your valuable feedback and suggestions on our revised paper. We appreciate your comments regarding the need for shorter sentences and suggesting proposals for future research in the conclusion. To address your concerns, we have shortened the sentences in our manuscript to improve readability and comprehension. Additionally, we have added proposals for future research in the conclusion section to provide directions for further investigation. Once again, many thanks.

Reviewer 2 Report

Dear author/s,

Thank you for the opportunity to review this manuscript. I hope you find my comments useful as you consider revising the paper. The topic is fitting with the aim and scope of the Journal. I hope this review provides some useful feedback and wish you the best of luck with the development of this paper!

Additional Questions:

1.      Originality:  Does the paper contain new and significant information adequate to justify publication?: Yes. he authors identified an interesting topic and provided significant information on the relation of Transformational Leadership and Innovative Behavior and Performance in the Imposed Change: The Roles of Commitment to Change and Organizational Support for Creativity”. However, there are some issues that need clarification. More importantly, the discussion regarding the research gaps is missing!

2.      Relationship to Literature:  Does the paper demonstrate an adequate understanding of the relevant literature in the field and cite an appropriate range of literature sources?  Is any significant work ignored?: The literature review section need improvement. You need to cite the below articles regarding leadership.

Aminah, H., Lin, P.-K., Susita, D., Helexandra, L., & Moslehpour, M. (2022). How does servant leadership affect public employees organizational citizenship behavior? The mediation effect of organizational culture and knowledge sharing. International Journal of eBusiness and eGovernment Studies, 14(1), 361-387.

Santoso, I. H., Wanto, H. S., & Siswati, E. (2021). The influence of transformational leadership and innovation on organizational and individual outcomes evidence from Indonesia. International Journal of eBusiness and eGovernment Studies, 13(2), 17-32.

3.      Methodology:  Is the paper's argument built on an appropriate base of theory, concepts, or other ideas?  Has the research or equivalent intellectual work on which the paper is based been well designed?  Are the methods employed appropriate?: The paper is well structured and follows the standards.

4. Results:  Are results presented clearly and analysed appropriately?  Do the conclusions adequately tie together the other elements of the paper?: The results section need to improve.

5. Implications for research, practice and/or society:  Does the paper identify clearly any implications for research, practice and/or society?  Does the paper bridge the gap between theory and practice? How can the research be used in practice (economic and commercial impact), in teaching, to influence public policy, in research (contributing to the body of knowledge)?  What is the impact upon society (influencing public attitudes, affecting quality of life)?  Are these implications consistent with the findings and conclusions of the paper?: The implications part can improve.

6. Quality of Communication:  Does the paper clearly express its case, measured against the technical language of the field and the expected knowledge of the journal's readership?  Has attention been paid to the clarity of expression and readability, such as sentence structure, jargon use, acronyms, etc.: A professional review of the language is strongly suggested because several parts of the text are unclear.

Author Response

You can post and reply to comments about the manuscript here. Note that the reviewers can also read these comments.

: We want to first thank you and the Reviewer for providing us with the opportunity to revise and resubmit our research. It is clear that you and the Reviewers gave this manuscript (2277882) very careful attention, and we have tried to reciprocate by addressing all of the various comments to the fullest extent possible. Below we have reproduced reviewers’ feedback in bold italics, followed by our response (blue color) in regular font. We carefully read the three reviewers’ comments and addressed and/or responded to each comment. We greatly appreciate this opportunity to revise and resubmit our manuscript. We welcome your continued feedback as we strive to make our manuscript as helpful to the field as possible.

Reviewer: 2

Recommendation:

Thank you for the opportunity to review this manuscript. I hope you find my comments useful as you consider revising the paper. The topic is fitting with the aim and scope of the Journal. I hope this review provides some useful feedback and wish you the best of luck with the development of this paper!

Comments:

  1. There are some issues that need clarification. More importantly, the discussion regarding the research gaps is missing.

Response to reviewer: We greatly appreciate your suggestion. Regarding the research gaps, we agree that we did not explicitly discuss them in the paper. We apologize for this oversight and have now included them in introduction section and added research question to enhance the clarity. Once again, many thanks.

  1. The literature review section need improvement. You need to cite the below articles regarding leadership.

Aminah, H., Lin, P.-K., Susita, D., Helexandra, L., & Moslehpour, M. (2022). How does servant leadership affect public employees organizational citizenship behavior? The mediation effect of organizational culture and knowledge sharing. International Journal of eBusiness and eGovernment Studies, 14(1), 361-387.

Santoso, I. H., Wanto, H. S., & Siswati, E. (2021). The influence of transformational leadership and innovation on organizational and individual outcomes evidence from Indonesia. International Journal of eBusiness and eGovernment Studies, 13(2), 17-32.

Response to reviewer: Thanks for your comment. Based on your suggestion, we have cited the two articles in the paper to improve the section. Once again, many thanks.

  1. The results section need to improve.

Response to reviewer: We appreciate your constructive comments and suggestions, and we have made revisions accordingly. In response to your comment about the results section, we have added an interpretation of the statistical analysis to provide a better understanding of our findings. We hope that this addition clarifies the results and their significance.

  1. The implications part can improve.

Response to reviewer: We appreciate your constructive criticism and suggestions to improve the implications section of our paper. We have revised the implications section to provide a more comprehensive explanation of both theoretical and managerial contributions of our study. Regarding the theoretical implications, we have included a detailed discussion of how our study contributes to the transformational leadership, change, and innovation literature. In terms of managerial implications, we have expanded our discussion on the practical implications of our study. We emphasized the importance of transformational leadership in promoting employee innovation and how organizations can enhance employee innovation through transformational leadership.

  1. A professional review of the language is strongly suggested because several parts of the text are unclear.

Response to reviewer: We appreciate your suggestion regarding the language used in our paper. We have taken your comments seriously and used a professional proofreading service to ensure the clarity and coherence of the text. We understand the importance of clear and concise language in academic writing, and we apologize for any confusion caused by unclear parts of the text. We have carefully reviewed the manuscript and made the necessary revisions to improve its readability and comprehension.

Reviewer 3 Report

This is an interesting study investigating the relationship between transformational leadership and employee innovation (in terms of behavior and performance). More specifically, this relationship is analyzed with the les of whether one could find a mediating effect between the two variables in employee commitment to change. Then, the impact of employee commitment to change on employee innovation was further dissected by investigating whether organizational support for creativity would moderate the link. The authors tell us that moth the mediator as well as the moderator appears to be positively associated with the said variables. The study deals with an important field of ongoing investigation and makes a noteworthy contribution. There are several things that need to be considered:

·        The abstract is extremely dense and difficult to read. It takes a while until one understands which relationships are being analyzed. However, the idea of the abstract is to give a very fast overview about the study. The abstract could be improved by making clear which variables are interesting and which relationships are being analyzed. For example, something in the direction of: We were interested in the relationship between A and B. First, we analyzed whether A and B was mediated by a third variable X. Results showed that this was the case. Second, we wanted to know whether the relationship between X and B was moderated by a fourth variable Y. Results also showed that this was the case.

·        It seems like there might have been a word missing in the sentence on line 77: “Overall continue to have relatively scant evidence for these…”

·        On line 83, the authors write of “This study”. I assume you mean the study you are about to publish right now? If this is the case, please write something like “The present study” or “The current study”, otherwise – since you are still in the literature section – it is not a hundred percent clear whether you are referring to a study you just cited or the present study.

·        The methodological approach seems sensible but the paper looks somewhat chaotic – I am not sure if this was a copy-paste problem or whether the original document upon its submission looked different. Please make sure that the formatting is correct and makes more sense. For example, Figure 1 should be on the next page together with its title. Also with the measures on page 10 and 11. They are all written in bold letters, which makes them rather confusing.

·        There is a structural problem with the introduction and the literature review. I was expecting a literature review in the introduction but then there was another literature review that, at many points, reiterated many of the claims made in the introduction chapter. This is redundant. You can either treat it in the introduction OR then make a succinct introductory chapter that highlights the problem including the research question and then quickly summarizes what is about to come. Then you could use the literature review to formulate the hypotheses – but please avoid the redundancies in the introduction and the literature review.  

·        The graph in figure 1 is not how mediators are usually depicted. It would make more sense to have a triangle where A leads to B and has a further mediator on the top as C (you can look at a graph online, for example: https://bmcmedresmethodol.biomedcentral.com/articles/10.1186/s12874-021-01426-3)

·        The authors have generated a large dataset, which is very impressive. Well done.

·        The methods section is well described but it lacks an important feature. You need to describe in a further subchapter your statistical design, which is where you tell the readers which statistical analyses you actually perform and how you do that. You wait to do this until the methods section where you tell readers that you used SPSS and Andrew Hayes’ macros for mediation and moderation analyses (this is the standard plugin to do this). First, you have to put this information in the methods section and second, you have to credit Hayes that you used his script (you cite him at other places, but you should also credit the sources of the plugin).

·        There are interesting analyses being made but at times they are difficult or impossible to follow due to formatting errors. For example, the numbers in table two (at least the document I received) are cut off. Perhaps a squared page would make more sense for these kinds of tables.

·        It would be quite nice to have the results summarized in an illustration (for example, like this: https://www.researchgate.net/figure/Path-model-diagram-with-the-results-of-parallel-mediation-analysis-Path-model-showing_fig1_331748519)

·        It is a good idea, as it is done in the present paper, to split the discussion into a “theoretical implications” and a “managerial implications” subsection. Like this, it is immediately cleat where the theoretical and where the practical merits lie.

·        I don’t know if the authors added some supplementary materials (I don’t seem to have access to this). They should add supplementary materials with the detailed outputs of their PROCESS calculations.

Overall, this is a very interesting study with a substantial contribution to the literature and an interesting statistical approach. The paper is in strong need of revision, which mainly deals with its presentation. Hence, there needs to be a redaction, restructuring and perhaps also revisualization of the design as well as the results. If the authors do this, then I believe this has the potential to become a solid paper.

Author Response

You can post and reply to comments about the manuscript here. Note that the reviewers can also read these comments.

: We want to first thank you and the Reviewer for providing us with the opportunity to revise and resubmit our research. It is clear that you and the Reviewers gave this manuscript (2277882) very careful attention, and we have tried to reciprocate by addressing all of the various comments to the fullest extent possible. Below we have reproduced reviewers’ feedback in bold italics, followed by our response (blue color) in regular font. We carefully read the three reviewers’ comments and addressed and/or responded to each comment. We greatly appreciate this opportunity to revise and resubmit our manuscript. We welcome your continued feedback as we strive to make our manuscript as helpful to the field as possible.

Reviewer: 3

Recommendation:

This is an interesting study investigating the relationship between transformational leadership and employee innovation (in terms of behavior and performance). More specifically, this relationship is analyzed with the les of whether one could find a mediating effect between the two variables in employee commitment to change. Then, the impact of employee commitment to change on employee innovation was further dissected by investigating whether organizational support for creativity would moderate the link. The authors tell us that moth the mediator as well as the moderator appears to be positively associated with the said variables. The study deals with an important field of ongoing investigation and makes a noteworthy contribution. There are several things that need to be considered:

  1. The abstract is extremely dense and difficult to read. It takes a while until one understands which relationships are being analyzed. However, the idea of the abstract is to give a very fast overview about the study. The abstract could be improved by making clear which variables are interesting and which relationships are being analyzed. For example, something in the direction of: We were interested in the relationship between A and B. First, we analyzed whether A and B was mediated by a third variable X. Results showed that this was the case. Second, we wanted to know whether the relationship between X and B was moderated by a fourth variable Y. Results also showed that this was the case.

Response to reviewer: We appreciate your valuable feedback, which has helped us to improve the clarity and structure of our paper. We have carefully considered your suggestion regarding the abstract and have revised it accordingly. We now provide a clearer overview of our study, highlighting the main variables of interest and the analytical approach taken. Specifically, we sought to reflect your feedback in terms of structure. Once again, many thanks for your comments.

  1. It seems like there might have been a word missing in the sentence on line 77: “Overall continue to have relatively scant evidence for these…”

Response to reviewer: Thanks for your comment in detail. During the English proofreading process, there was a word missing in the sentence on line 77. We have revised the sentence by considering its meaning. Thanks.

  1. On line 83, the authors write of “This study”. I assume you mean the study you are about to publish right now? If this is the case, please write something like “The present study” or “The current study”, otherwise – since you are still in the literature section – it is not a hundred percent clear whether you are referring to a study you just cited or the present study.

Response to reviewer: Thanks for your feedback on clarification of the sentence. We have revised the work from ‘this study’ into ‘the present study’. Many thanks.

  1. The methodological approach seems sensible but the paper looks somewhat chaotic – I am not sure if this was a copy-paste problem or whether the original document upon its submission looked different. Please make sure that the formatting is correct and makes more sense. For example, Figure 1 should be on the next page together with its title. Also with the measures on page 10 and 11. They are all written in bold letters, which makes them rather confusing.

Response to reviewer: Thanks for your feedback in detail. Once we submitted the paper to the journal, there might be problem with a copy-paste by the system. According to your feedback, we manually changed the format and make it more sensible to the readers. Once again, many thanks.

  1. There is a structural problem with the introduction and the literature review. I was expecting a literature review in the introduction but then there was another literature review that, at many points, reiterated many of the claims made in the introduction chapter. This is redundant. You can either treat it in the introduction OR then make a succinct introductory chapter that highlights the problem including the research question and then quickly summarizes what is about to come. Then you could use the literature review to formulate the hypotheses – but please avoid the redundancies in the introduction and the literature review.  

Response to reviewer: Thank you for your valuable feedback regarding the structural problem with our introduction and literature review. We apologize for the redundancies and lack of clarity in these sections. We appreciate your suggestions for improvement, and we have made revisions to address your concerns. As per your suggestion, we have revised the introduction to include a succinct summary of the problem and research question. We have eliminated the redundancies and kept the literature review focused on formulating hypotheses. We believe that this approach will make the introduction and literature review more concise and clear.

  1. The graph in figure 1 is not how mediators are usually depicted. It would make more sense to have a triangle where A leads to B and has a further mediator on the top as C (you can look at a graph online, for example: https://bmcmedresmethodol.biomedcentral.com/articles/10.1186/s12874-021-01426-3).  

Response to reviewer: Thanks for your valuable feedback. We have revised the research model (Figure 1) as you requested. Once again, many thanks.

  1. The authors have generated a large dataset, which is very impressive. Well done.  

Response to reviewer: Thanks for your complimentary feedback.

  1. The methods section is well described but it lacks an important feature. You need to describe in a further subchapter your statistical design, which is where you tell the readers which statistical analyses you actually perform and how you do that. You wait to do this until the methods section where you tell readers that you used SPSS and Andrew Hayes’ macros for mediation and moderation analyses (this is the standard plugin to do this). First, you have to put this information in the methods section and second, you have to credit Hayes that you used his script (you cite him at other places, but you should also credit the sources of the plugin).  

Response to reviewer: We acknowledge the importance of describing our statistical design in a further subchapter to provide readers with a clear understanding of the analyses performed in our study. As suggested, we will include a subchapter in the methods section to provide detailed information on our statistical design. Specifically, we will explain the statistical analyses conducted in our study and how they were performed, including the use of SPSS and Andrew Hayes’ macros for mediation and moderation analyses. Furthermore, we will credit Hayes for the use of his macros and reference the appropriate sources.

  1. There are interesting analyses being made but at times they are difficult or impossible to follow due to formatting errors. For example, the numbers in table two (at least the document I received) are cut off. Perhaps a squared page would make more sense for these kinds of tables.

Response to reviewer: Thanks for your feedback and sorry for any unconvincing your review. I have shortened the variable name and make it easy for the readers to look at the table. Once again, thanks.

  1. It would be quite nice to have the results summarized in an illustration (for example, like this: https://www.researchgate.net/figure/Path-model-diagram-with-the-results-of-parallel-mediation-analysis-Path-model-showing_fig1_331748519).

Response to reviewer: Thanks for your valuable feedback. We added figure 5 to illustrate the summarized results for readers to easily identify the results. Many thanks.

  1. It is a good idea, as it is done in the present paper, to split the discussion into a “theoretical implications” and a “managerial implications” subsection. Like this, it is immediately cleat where the theoretical and where the practical merits lie.

Response to reviewer: Thanks for your complimentary feedback.

  1. I don’t know if the authors added some supplementary materials (I don’t seem to have access to this). They should add supplementary materials with the detailed outputs of their PROCESS calculations.

Response to reviewer: Thanks for your valuable comment. We have revised the tables (Table 6 and 7) including outputs of PROCESS macro analysis and cited an article (Kang et al., 2022) employing the same method in Behavioral Sciences. Once again, many thanks.

Reference

Kang, H.; Song, M.; Li, Y. Self-Leadership and Innovative Behavior: Mediation of Informal Learning and Moderation of Social Capital. Behav. Sci. 2022, 12, 443. https://doi.org/10.3390/bs12110443

  1. Overall, this is a very interesting study with a substantial contribution to the literature and an interesting statistical approach. The paper is in strong need of revision, which mainly deals with its presentation. Hence, there needs to be a redaction, restructuring and perhaps also revisualization of the design as well as the results. If the authors do this, then I believe this has the potential to become a solid paper.

Response to reviewer: We appreciate your positive comments about the potential contribution of our study to the literature and the statistical approach we employed. We understand that the paper needs significant revision, particularly in terms of its presentation, including redaction, restructuring, and revisualization of the design and results. We agree with your suggestions and will work on revising the manuscript accordingly. We will pay careful attention to the structure and organization of the paper, ensuring that it is clear and concise. We will also consider the use of visual aids to help convey our findings more effectively. Again, we appreciate your constructive feedback and will take it into account as we revise the paper. We hope that the revised manuscript will meet the standards of the journal and look forward to the opportunity to resubmit it for further review.

Round 2

Reviewer 1 Report

Dear authors,

This version answered all my concerns.

Author Response

Thank you

Reviewer 3 Report

The authors have made all the necessary changes to the manuscript. Many thanks.

Author Response

Thank you